# Exploring the Link between Smartphone Use, Recorded Violence, and Social Sharing in 80 Case Studies in Italy

Amelia Rizzo [1,2,*] , Emanuela Princiotta [3] and Giada Iuele [4]

1. Clinical Psychology, Social Psychology and Organizational Psychology, Department of Clinical and Experimental Medicine, University of Messina, 98122 Messina, Italy
2. Department of Cognitive, Psychological, Pedagogical Sciences and Cultural Studies, University of Messina, 98122 Messina, Italy
3. Clinical and Preventive Psychological Sciences and Techniques, Department of Clinical and Experimental Medicine, University of Messina, 98122 Messina, Italy
4. Clinical and Health Psychology in the Life Cycle, Department of Clinical and Experimental Medicine, University of Messina, 98122 Messina, Italy; giadaiuele@gmail.com
* Correspondence: amrizzo@unime.it

**Abstract:** The increasing prevalence of violence recorded and shared through smartphones in today's digital age has raised concerns about the underlying reasons driving such behavior. However, the lack of experimental studies and scientific evidence exploring the relationship between smartphone use and acts of violence has hindered our understanding of this phenomenon. To bridge this gap, the present study aimed to investigate the potential link between smartphone usage and the perpetration of violence, specifically focusing on incidents where violent acts were recorded and shared publicly. Given the challenges associated with directly observing such occurrences and the limitations of self-reporting due to social desirability bias, the study adopted a novel approach by analyzing major news outlets. Cross-referencing the most recent cases involving 80 episodes of violence, spanning from 2017 to 2023, accompanied by smartphone-recorded videos, the research aimed to gain insights into the role and outcomes of content dissemination. The findings revealed a concerning trend, indicating a rise in violence perpetrated with the aid of smartphones, where subsequent sharing on social networks and instant messaging platforms contributed to the viral spread of such content. This study provides valuable insights into the connection between smartphone usage, violence, and the sharing of violent content. The implications of these findings highlight the need for further research and the development of tools to detect and address violence-related issues in the digital space. Moreover, it emphasizes the importance of responsible social media usage and collective efforts to curb the spread of violent content and foster a safer online environment.

**Keywords:** smartphone use; violence; video; social sharing

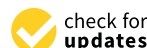



## 1. Introduction

Violence and aggression are intriguing subjects within the realm of social psychology, a branch of psychology that focuses on understanding how an individual's environment, exhibited behaviors, and interpersonal relationships interact [1]. In today's digital age, the prevalence of smartphones and the internet has extended the scope of violence, providing new avenues for its expression [2]. The combination of social psychology and technology highlights the need to examine the intricate interplay between human behavior and smartphone-mediated violence, fostering a deeper comprehension of the evolving complexities of human interactions in the digital era [3].

Delineating the construct of aggression from that of violence is essential. Psychoanalytically, aggression is viewed as a tendency—or a set of tendencies—expressed through actual or fantasized behavior aimed at harming, demolishing, coercing, or humiliating another person [4].

From a cognitive behavioral standpoint, aggression is seen as a form of non-assertive interaction. It involves imposing one's thoughts, beliefs, emotions, and goals on others, often at their expense. This behavior is rooted in self-aggrandizement and the devaluation of others. It sanctions achieving personal goals by any means, without regard for others, and may include manipulative and violent acts. Typically, aggressive individuals lack the cognitive ability to foresee the long-term harmful effects of their actions. Verbally, this is reflected in dogmatic and absolutist communication, sometimes deceitful or exaggerated, characterized by offensive and provocative language. Non-verbally, it manifests through a steady and intense gaze, loud and abrupt speech, an intimidating posture, and threatening gestures [5].

On the other hand, the term violence stems from the Latin "violentus", meaning "violent", with the root "vis-" implying "force" [6]. The World Health Organization [7] defines violence as the intentional use of physical force or power, actual or threatened, against oneself, another person, a group, or a community, leading to injury, psychological harm, maldevelopment, deprivation, or death. The United Nations' [8] description aligns with this, framing violence as any action that causes or has the potential to cause physical, sexual, or psychological harm. This includes threats of such acts, coercion, and arbitrary deprivation of liberty, in public or private life.

Thus, it is evident that violence, whether physical, verbal, or through media, inherently involves the use of force and the intent to compromise others' autonomy.

In modern media, violence manifests in diverse and impactful ways:

"Entertaining" Violence: Commonly seen in cartoons and videos, this form features protagonists engaging in perilous feats with no real repercussions, presenting danger as more of an adventurous element rather than a serious threat [9].

Beating Videos: (such as smack cam, slap cam, happy slapping): This trend is especially prevalent among teenagers who record and share videos of themselves being physically harmed. These videos often blur the line between reality and performance, leading to both physical harm and emotional distress due to public humiliation [10].

Extremist Violence: This involves posts, videos, and music that carry propaganda from extreme ideological groups, ranging from far-right and far-left to various religious ideologies [11].

Incitement to Hatred: Often found on social media, instant messaging, forums, and video platforms, this category includes posts and comments that promote discrimination and hostility against certain individuals or groups [12].

Violent Games: With advancements in technology, violence in online games has become increasingly realistic, portraying aggressive scenarios in a graphic and immersive manner [13].

Violent Films: The film industry, particularly in genres like thrillers, action movies, war films, and horror, frequently incorporates violence, impacting the young audience notably [14].

Violent Songs and Music Videos: Music, a crucial part of youth culture, often includes violent themes in genres like rap, heavy metal, punk, and goth. These songs and videos can both directly and indirectly incite violence.

Violent Pornography: This category, disturbingly accessible online, includes hardcore content involving minors, non-consensual acts, and violence, sparking curiosity among teenagers [15].

Real Violence and Snuff Videos: Media also showcases actual violence from crimes, wars, terrorism, or natural disasters, often found in news outlets and newspapers, exposing viewers to the harsh realities of the world [16].

Additionally, there are circulating videos, such as snuff videos containing scenes of murders (though it is unclear if they are real). Islamic groups constantly publish films of executions and torture. Self-inflicted Violence: Due to the opportunities provided by digital media, self-harming behavior among young people (e.g., cutting the skin) has reached a new level. They also use forums and instant messaging services to incite each other in the

case of anorexia, bulimia [17], or suicidal thoughts [18]. The violence perpetrated with the help of the smartphone is increasing and can easily spread virally nowadays. These videos are posted on social networks and shared on instant messaging apps and platforms [19]. Unfortunately, these actions are often reinforced by hundreds of supporters who, in some cases, incite the perpetrator of the violence to continue [20].

Understanding the intricacies of violence in the digital age, particularly smartphone-mediated violence, presents researchers with formidable hurdles. Two predominant methods for studying this phenomenon, direct observation and self-reporting with direct questions, encounter significant obstacles [21]. Direct observation proves challenging due to ethical constraints and the impracticality of witnessing real-time violent acts. On the other hand, self-reporting is marred by the social desirability bias, where participants may alter responses to conform to societal norms or avoid judgment. Additionally, the scarcity of effective tools for detecting violence and the limited success of explicit measures further hinder comprehensive data collection and analysis [22]. In light of these challenges, researchers are compelled to devise innovative and ethical methodologies to unveil the enigmatic nature of smartphone-mediated violence. By overcoming these research barriers, we can foster a deeper comprehension of the evolving landscape of violence, enabling the development of evidence-based strategies to address this pressing issue in today's technologically interconnected world.

Given the lack of experimental studies and scientific evidence linking the use of smartphones to acts of violence, with concomitant recording and sharing of content in public, a pilot study is necessary in order to identify what may be the main reasons underlying the act of recording violence.

Based on the aforementioned challenges, the general aim of this study was to investigate the connection between smartphone-mediated violence and its subsequent sharing via major news outlets. To achieve this objective, the research adopted a specific method of analyzing 80 recent cases of violence reported in Italy between 2017 and 2023.

By cross-referencing incidents where violence was explicitly captured and shared through smartphone-recorded videos, the study specifically aimed to categorize uses and motivations and analyze co-occurrence of variables.

## 2. Materials and Methods

An internet-based search was conducted by E.P. and A.R. and validated by G.I. on major Italian newspapers using keywords such as "violence recorded with smartphones" and combinations of words like "beat/beating", "assault/assaulting", "violence/violating", "threaten/threatening", "injure/injuring", "insult/insulting", "film everything with smartphone", "share video", "spread video", "post everything on social media", and "video ends up on social media". News about violence without a clear smartphone role was excluded. When possible, all available information about the episode was printed, cross-compared with different sources, and verified, raising the level of reliability of the information collected and allowing a realistic outlook to be obtained in terms of quantity and quality. The cases were organized in an Excel sheet, where the first column indicates the place and date of the event, the second column summarizes the reported episode of violence, and the third column represents a classification of the main and secondary motivation behind smartphone use and video use.

### 2.1. Ethics

In the present study concerning episodes of violence and the role of smartphones, we have strictly adhered to ethical guidelines to protect the privacy and confidentiality of the individuals involved. All victim names and personal information have been anonymized to prevent any possibility of identification. We have implemented rigorous data encryption and access control measures to ensure that sensitive information remains secure.

*2.2. Analysis Tools*

A Microsoft Excel sheet served as the initial repository for the systematic organization of quantitative data, which included a range of variables from survey responses to incident logs. For qualitative insights, the software MAXQDA Analytics Pro 2022 (Release 22.7.0) was instrumental. It facilitated a rigorous segment analysis through which a coding schema was developed, categorizing the data into discrete phenomena reflecting varying manifestations of violence in the context of smartphone usage. This coding process was rooted in grounded theory, allowing for emergent themes to be quantified and compared [23]. The analytical prowess of MAXQDA was further leveraged to conduct a co-occurrence analysis. This advanced statistical method permitted an exploration of the relationships between the established codes, shedding light on the prevalence and context of violent incidents correlated with smartphone use. Through this multifaceted analytical approach, the research was able to offer a data-driven understanding of the patterns and contextual factors that underpin the nexus of violence and smartphone use.

## 3. Results

The following Table 1 presents the results of the search, with 80 cases of violence involving the use of smartphones in Italy.

**Table 1.** Case studies (*n* = 80) of smartphone-related violence.

| Location and Date | Violence Episode | Smartphone Role | Secondary Role |
|---|---|---|---|
| Acireale, 27 December 2022 | A woman was beaten for months and recorded and threatened during intimate relations. | Blackmail | |
| Ancona, 10 May 2022 | A girl took her phone for repair, after which threats to publish intimate videos involving her and her boyfriend were made. | Blackmail | |
| Anzio, 7 May 2021 | A 28-year-old boy was beaten and forced to undress then recorded with a smartphone. | Virality | Humiliation |
| Artena, 9 November 2022 | After the murder of a boy, videos involving the torture of birds and sheep were found on the phones of the accused. | Perversion | |
| Bari, 20 October 2022 | Sexual violence on WhatsApp in the Bari area. | Humiliation | Perversion |
| Belluno, 23 May 2023 | Two girls are instigated to fight by peers. The graphic imagery was spread around chats for days in the form of a viral video. | Virality | Incitement to Violence |
| Bologna, 21 November 2022 | An adult man and a very young individual were involved in a brawl in a municipal parking lot, while a large group of boys and girls filmed the gruesome scene, limiting themselves to commenting without intervening. | Virality | Bystander Effect |
| Bologna, 14 January 2017 | The perpetrator blackmailed his ex-girlfriend. | Blackmail | |
| Bolzano, 27 January 2023 | A group sexual assault was filmed with phones. | Humiliation | Perversion |
| Brescia, 21 September 2021 | A scene was filmed with cell phones by young bullies, who then spread the video on the internet in Instagram stories and WhatsApp chats to resonate the act and ridicule the female victim. | Incitement to Violence | Blackmail/ humiliation |
| Camposano, 1 September 2022 | Shopping in Nolano ended with insults and violence as two women fought and other customers filmed them. Views: 50,532 | Virality | Bystander Effect |
| Caserta, 24 July 2022 | Minors between the ages of 13 and 15 fought and were captured by friends' smartphones in a fifteen-second video that went viral. | Virality | Incitement to Violence |
| Castellammare, 27 May 2018 | A 12-year-old was raped by four boys. She reported having been filmed by a smartphone. | Blackmail | Humiliation/ perversion |
| Catania, 10 December 2021 | The victim was abused and blackmailed with a video made using a smartphone. | Blackmail | |
| Catania, 15 February 2020 | Students were beaten, and the video was put online. | Blackmail | Humiliation |
| Cefalù, 29 November 2019 | A 14-year-old with mild disability was mocked and attacked. Three reports were filed. | Virality | Humiliation |
| Cerignola, 7 June 2023 | A fight among girls, with onlookers inciting violence and filming the scene with cell phones, the video of which was spread on WhatsApp. | Incitement to Violence | Virality |
| Cerignola, 9 June 2023 | Thirteen-year-olds fought in Cerignola. The incident was filmed in its entirety by children, and the video went viral. | Virality | |

**Table 1.** *Cont.*

| Location and Date | Violence Episode | Smartphone Role | Secondary Role |
|---|---|---|---|
| Chieti, 28 May 2023 | A boy was beaten two-on-one at a municipal while filmed by friends. Videos were posted on Telegram. | Virality | |
| Chieti, 5 November 2022 | The subject killed his grandfather and filmed everything with a smartphone. | Perversion | Sociopathy |
| Cosenza, 31 August 2023 | Unprecedented violence took place on the Mirto Crosia waterfront: kicks, punches, and a video of the assault went viral among local WhatsApp groups. | Virality | |
| Empoli, 14 August 2019 | Three boys were raped in a car in an incident that was filmed with a smartphone. | Blackmail | Humiliation/ perversion |
| Florence, 17 July 2023 | Two people fought, and others watched and recorded with a mobile phone. | Bystander Effect | Virality |
| Florence, 18 May 2022 | A disabled person was beaten by a group of youths, and the video was posted on social networks. | Virality | |
| Florence, 2 August 2022 | Four men raped and filmed a woman in Florence. | Humiliation | Perversion |
| Fiuggi, 29 August 2023 | A young goat was kicked to death, and the incident was recorded with a smartphone, while friends laughed and encouraged the perpetrators. | Incitement to violence | |
| Foggia, 18 January 2023 | Used like a mannequin, the victim was raped and drugged by three men. | Perversion | |
| Frosinone, 22 October 2022 | A man forced his girlfriend to have sex with him by beating her while being threatened to publish the video online. | Blackmail | Humiliation |
| Genova, 28 February 2017 | A 12-year-old was beaten bloody and recorded with a cell phone; the images ended up on WhatsApp. | Virality | Spectacularization |
| Imperia, 22 November 2022 | A seagull was tortured on the promenade, beaten, and tied with a rope. All was captured on video and posted on social media. | Torture/cruelty | Virality |
| Latina, 19 January 2023 | A teacher was hit with a backpack by students. They filmed everything and posted the video on social media. | Challenge | Virality |
| Lecce, 12 April 2018 | A 10-year sentence was given to a man from Adrano for abusing a 12-year-old between 2016 and 2017. | Blackmail | Humiliation/ perversion |
| Licata, 26 January 2021 | Three disabled boys were beaten and ridiculed, and the entire event was recorded with cell phones. The victims were humiliated with paint on their faces and mocked on social media. | Virality | Humiliation |
| Lodi, 13 October 2023 | Girls fought while others cheered and recorded the scene in Lodi. The video, which went viral, was also shared by other children's parents. | Virality/ spectacularization | Evidence and complaint/ incitement to violence |
| Marano di Napoli, 9 February 2023 | An MP was attacked by two women in a shocking video. | Humiliation | |
| Messina, 10 June 2022 | The aggressor attacked their peer with kicks and punches due to jealousy. | Virality | Spectacularization |
| Milan, 27 September 2018 | Fighting and chaos among "Instagram Bullies" was shared on social networks. | Virality | |
| Milan, 14 November 2022 | A girl was assaulted in a Milan motel. Three individuals were arrested. | Blackmail | Humiliation/ perversion |
| Milan, 20 January 2023 | An American student was mistreated by two soccer players. | Blackmail | Humiliation/ perversion |
| Milan, 24 May 2023 | A woman was confronted by four officers. She was disturbing children. | Reporting | |
| Milan, 5 March 2022 | A security guard was attacked at McDonald's and filmed; four individuals were arrested. | Virality | |
| Modena, 26 January 2023 | A teacher was reported for inappropriate conduct and recorded by a student. | Reporting | |
| Montefusco, 22 May 2023 | An individual killed a kitten and posted a video on TikTok. | Perversion | Sociopathy |
| Naples, 9 September 2022 | A 13-year-old girl was attacked, and the incident was filmed with a cellphone. After threats, the attackers began to push and strike her. A girl recorded everything, intending to send it to friends or possibly post it on social networks. | Blackmail/ humiliation | Virality |
| Naples, 2 December 2022 | A person with a disability was attacked for likes. | Virality | Humiliation |

**Table 1.** *Cont.*

| Location and Date | Violence Episode | Smartphone Role | Secondary Role |
|---|---|---|---|
| Orzinuovi, 20 September 2022 | After school, two underage girls fought, encouraged by a crowd that filmed with smartphones. The footage spread among students' and teachers' phones. | Virality | Incitement to violence |
| Padua, 10 October 2022 | A person became intoxicated and fainted and was then assaulted and filmed by friends. The video circulated among friends and acquaintances. | Blackmail/ humiliation | |
| Perugia, 1 July 2023 | Two girls fought in the street while friends laughed and filmed the scene. The video ended up online. | Virality | Incitement to violence |
| Perugia, 13 May 2022 | Attackers kicked and punched a girl and filmed everything. The 14-year-old was hospitalized. | Virality | Humiliation |
| Pompeii, 16 October 2022 | A child was in tears after being hit by bullies. | Humiliation | |
| Pontedera, 3 November 2022 | A video by students in a high school in Pontedera captured a teacher hitting a student who was mocking him. | Reporting | |
| Ragusa, 30 March 2017 | A young individual was forced to dance undressed at a bus stop. | Virality | Humiliation |
| Ravenna, 27 October 2017 | An assault on an unconscious 19-year-old from Ravenna was filmed with a smartphone. Two were arrested. | Blackmail | Humiliation/ perversion |
| Reggio Calabria, 1 August 2022 | A twenty-against-one brawl at night resulted in the victim being brutally beaten. | Virality | Spectacularization |
| Rimini, 31 January 2023 | In Rimini, individuals were filmed mistreating their dog. | Reporting | |
| Rodi Garganico, 24 September 2023 | Four boys brutally attacked a duck, recording and sharing the act on social media, contributing to a disturbing trend of animal violence. | Virality | Torture/cruelty |
| Rome (Cinecittà), 4 February 2021 | Accused of being a snitch, a boy was attacked by seven classmates. | Blackmail | Humiliation |
| Rome, 10 March 2022 | A child admitted to his mother that he used drugs with six young people. He was forced to submit after being attacked by a group of his peers, who recorded everything with a cellphone and spread the video on social networks. He also contemplated suicide. | Blackmail/ humiliation | Virality |
| Rome, 15 October 2022 | Inappropriate content involving a 2-year-old was recorded with a smartphone. | Humiliation | Perversion |
| Rome, 2 May 2021 | A 17-year-old boy was severely beaten by a group of individuals; the attack was recorded with a smartphone. | Virality | Humiliation |
| Rome, 30 June 2021 | A same-sex couple who kissed on a bus were attacked. | Humiliation | Virality |
| Rome, 6 December 2017 | An assault in the center of Rome was filmed with a smartphone. Three were arrested. | Blackmail | Humiliation/ perversion |
| Rome, 7 October 2022 | A 14-year-old girl was slapped in class; classmates recorded everything, but no one intervened. | Virality | Humiliation |
| Rovigo, 26 October 2022 | Students shot at a teacher with a pellet gun. | Virality | Humiliation |
| Rutigliano, 29 May 2021 | A group of boys attacked a peer with kicks and punches, recording the violence and posting the video on TikTok. The video went viral. | Virality | |
| Salerno, 4 February 2021 | Bullies at school attacked a girl with a disability. The whole scene was captured and spread among the children. | Virality | |
| San Felice Circeo, 11 August 2021 | Youngsters fought in an amphitheater and recorded themselves with smartphones. | Incitement to violence | |
| Santa Maria a Vico, 2 February 2018 | A teacher's face was scarred by a student, with 32 stitches required. | Humiliation | |
| Sarno, 30 March 2023 | Individuals attacked a homeless person with disabilities and posted the video. | Virality | |
| Sassari, 9 February 2022 | A young person was assaulted with a knuckleduster, and the attacker's friends recorded everything. The two minors were placed in a community home. No one intervened. | Bystander effect | Virality |
| Scalea, 31 January 2023 | Video of an assault circulated on the Web. | Virality | Spectacularization |
| Siena, 30–31 May 2021 | A soccer player assaulted a 21-year-old girl. | Perversion | |
| Spalato, 20 April 2020 | A clandestine mass was held on Catholic Easter day, contrary to the prohibition imposed by the coronavirus epidemic. Journalists documenting the event were assaulted and insulted. | Evidence and report | |
| Taranto, 26 May 2022 | The victim was threatened over provocative photos. | Blackmail | |
| Terracina (Lazio), 27 October 2021 | Group violence was recorded: one suspect denied accusations, while the other did not respond. | Blackmail | Perversion |

**Table 1.** *Cont.*

| Location and Date | Violence Episode | Smartphone Role | Secondary Role |
|---|---|---|---|
| Todi (Umbria), 23 January 2023 | A 17-year-old was assaulted and filmed in a shelter. | Humiliation | Perversion |
| Torre Annunziata (NA), 24 June 2021 | The victim was attacked for being homosexual by seven perpetrators. | Humiliation | Virality |
| Vicenza, 2 October 2018 | Students attacked a teacher. | Blackmail | Humiliation |
| Viterbo, 29 April 2019 | A local political figure and a military member attacked and assaulted a woman; one was arrested. | Humiliation | Perversion |
| Vittoria (RG), 19 August 2022 | A young individual's hair was cut off in a filmed incident. | Humiliation | |

The collection of cases provided paints a distressing picture of various acts of violence, harassment, and cruelty that have occurred across Italy. These events span from 2017 to 2023 and are scattered throughout different regions, including Northern, Central, and Southern Italy and the islands (See Figure 1).

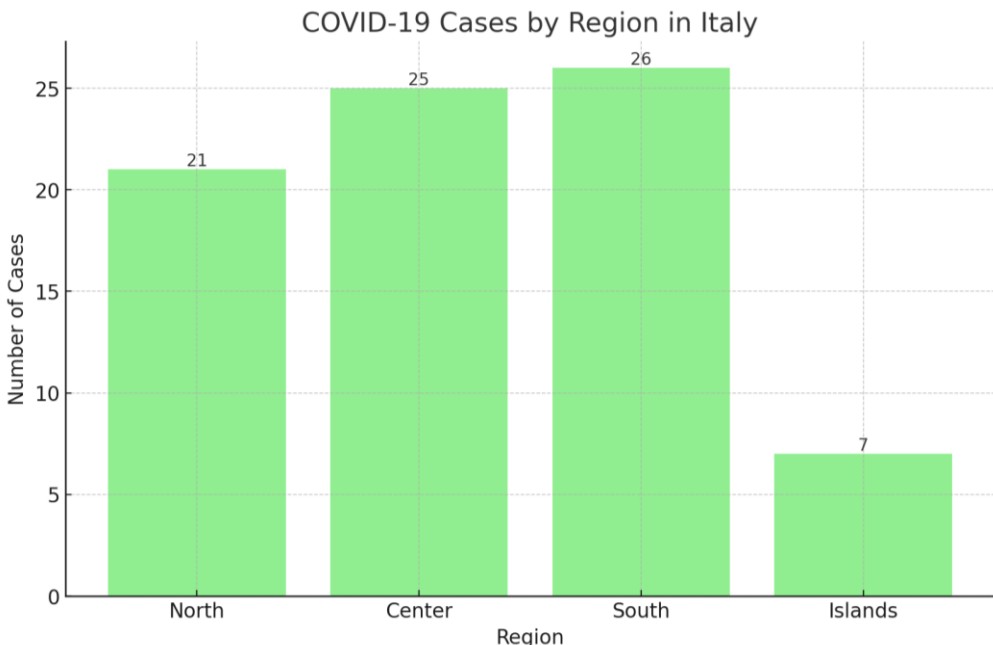

**Figure 1.** Number of cases per region.

The incidents range from physical assaults to sexual violence, bullying, harassment, animal abuse, and even instances of child pornography. These actions represent a wide spectrum of harmful behaviors. The victims in these incidents include children, teenagers, adults, and even animals. The perpetrators, on the other hand, include peers, classmates, students, teachers, strangers, and even law enforcement officers. Both male and female individuals are involved as victims and perpetrators.

The common theme is the use of smartphones to record these incidents. Perpetrators and bystanders often capture these events on their phones, and in some cases, videos are shared on social media platforms, which can lead to the incidents going viral. Technology, particularly smartphones, plays a significant role in these incidents. Many of the events were captured on mobile devices, often shared on social media platforms, and sometimes even used as tools of intimidation or blackmail. Videos and images are shared on social media platforms like Instagram, WhatsApp, TikTok, and more. This amplifies the reach and impact of these incidents, leading to public outrage and potential legal consequences for the perpetrators.

Through a qualitative analysis of the reported cases, different motivations were identified (See Figure 2):

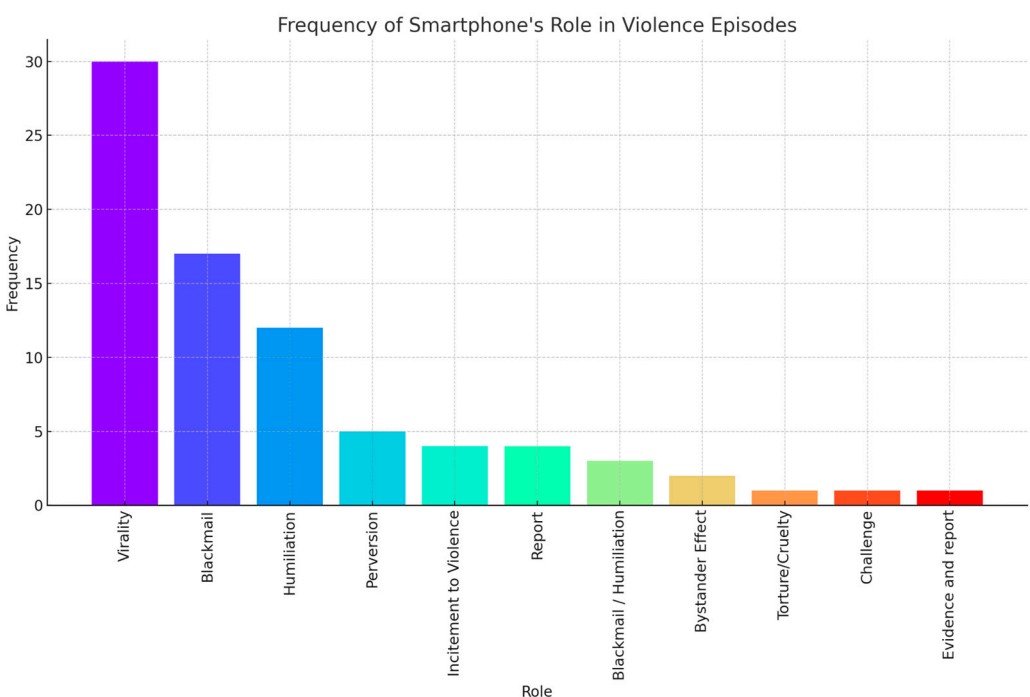

**Figure 2.** Frequency of the role of smartphones in violence episodes.

Virality: Smartphones enable the rapid spread of content through social media and messaging apps. When violent incidents are recorded and shared, they can go viral, reaching large audiences quickly. This not only extends the impact of the violent act beyond the immediate location and moment but also can lead to widespread notoriety for the perpetrators and additional trauma for the victims.

Blackmail: The ability to capture and store images and videos on smartphones makes it possible for individuals to use this content as leverage over others. Sensitive or compromising material can be threatened to be shared publicly unless the victim complies with certain demands, thereby using the smartphone as a tool for coercion.

Humiliation: The act of recording a violent incident and sharing it can be used to deliberately humiliate the victim. The permanence and replicability of digital content mean that once a humiliating video is shared, it can be difficult or impossible to remove from the public domain, resulting in prolonged and repeated trauma for the victim.

Bystander Effect: The presence of smartphones can sometimes lead to a phenomenon where witnesses to violence are more likely to record the event than to intervene or help the victim. This "bystander effect" can be compounded by the prospect of capturing footage that gains attention or goes viral.

Incitement to Violence: Smartphones can be used to incite or encourage violence by allowing individuals to communicate plans, share violent ideas, or call for others to join in on violent actions. This can be through direct messages, social media posts, or through forums and chat groups.

Perversion: In some cases, smartphones can be involved in perverse acts, such as the recording or dissemination of sexually explicit or abusive material without consent. This can include cases of sexual exploitation or abuse that are facilitated or documented using the device.

Reporting: On a more positive note, smartphones can also be used to report violence, acting as a tool for victims or witnesses to document incidents and seek help. This can involve recording evidence of violence to support legal action or using communication features to call for immediate assistance.

It is possible that these are not the only motivations or categories representing the purpose for which smartphones are used during violence. An example is represented by those who share recordings of violence to seek justice, for instance, to identify the perpetrators. This could lie on the boundary between using smartphones to gather evidence for reporting and sharing images on platforms to gain consensus.

The graph presented is a co-occurrence matrix that visualizes the relationship between different roles that smartphones play in episodes of violence. Each cell in the matrix represents the number of times two roles have been observed to occur together in incidents of violence (Figure 3).

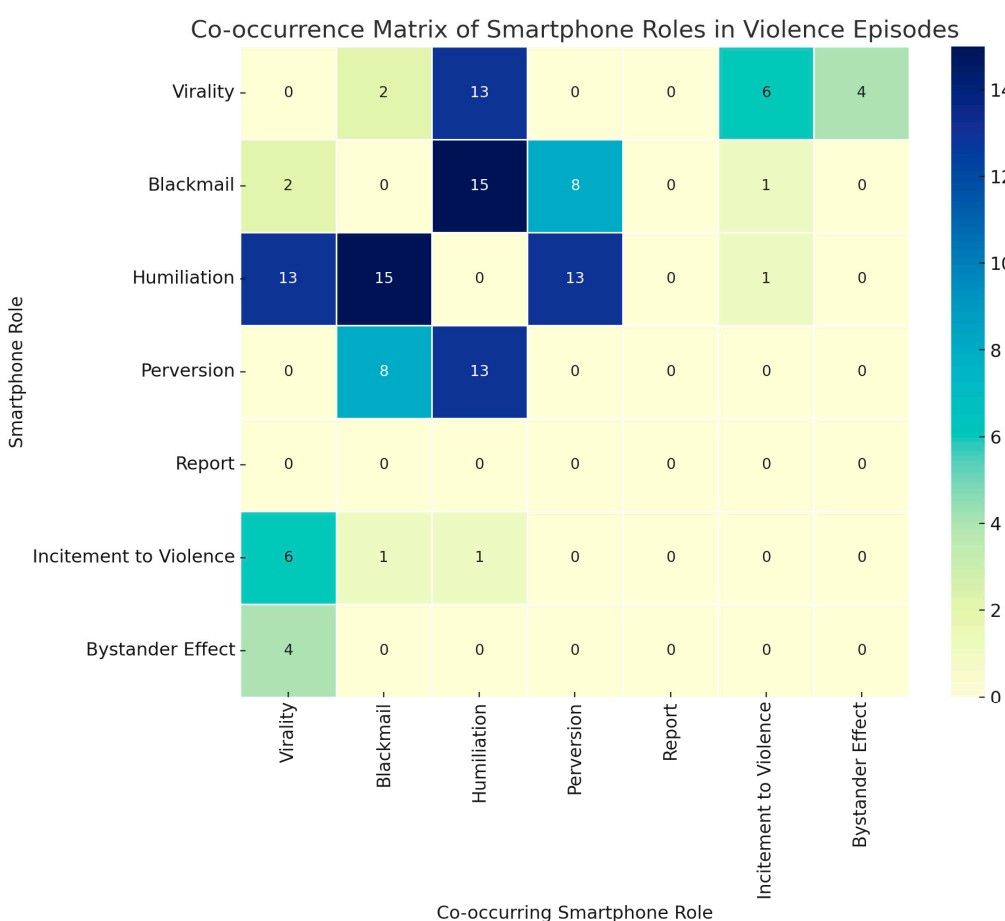

**Figure 3.** Co-occurrence matrix of smartphone roles in violence episodes. Note: a. The colors represent the frequency of co-occurrences, with lighter colors indicating higher frequencies. b. The numbers in the cells show the count of co-occurrences between the roles on the x and y axes.

A breakdown of the insights we can gather from the graph is given in the following.

Humiliation seems to be the role most commonly associated with smartphones in violent episodes, often co-occurring with virality (13 instances), blackmail (15 instances), and perversion (13 instances). This suggests that smartphones are frequently used to record and distribute content meant to embarrass or humiliate individuals.

Blackmail is another role with significant co-occurrence, especially with humiliation (15 instances) and perversion (13 instances). It indicates that incidents where smartphones are used for blackmail often involve humiliating or perverse content.

Virality has a notable connection with humiliation (13 instances) and to a lesser extent with incitement to violence (6 instances) and the bystander effect (4 instances). This highlights how episodes of violence captured on smartphones can rapidly spread online, potentially inciting further violence or causing bystander inaction.

Perversion is mostly linked with humiliation and blackmail, which might point towards the use of smartphones to perpetrate or share content of a sexual and exploitative nature.

Incitement to violence has a relatively lower but still noteworthy connection with virality, indicating that some violent episodes captured and shared via smartphones can lead to further incitement of violence.

The Bystander effect has the least co-occurrence with other roles, yet it still connects with virality, suggesting that in some cases, the widespread sharing of violent episodes leads to bystander apathy or inaction.

Reporting as a role has the least co-occurrence with other roles, which may imply that the use of smartphones to report violence is less common or less connected to other smartphone roles within violent episodes.

In other words, the gradient representing the co-occurrence of humiliation and blackmail suggests a disturbing trend where individuals use smartphones not only to shame victims but also to coerce them. The frequent co-occurrence of virality with other roles like humiliation and incitement to violence underscores the potent role of social media and the internet in spreading harmful content. This rapid dissemination can magnify the impact of violent acts.

## 4. Discussion

The primary aim of this study was to investigate the connection between smartphone-mediated violence and its subsequent sharing on social media. To achieve this objective, the research adopted a specific method of analyzing 80 recent cases of violence filmed and reported in the media. The relationship between violence and smartphone use is a complex and ever-evolving topic. On one hand, excessive smartphone use can lead to a range of problems, such as addiction [24], anxiety, depression [25], and social isolation [26], which may contribute to an increased likelihood of violent behaviors [27,28]. Furthermore, smartphone use can be associated with violent behaviors, such as cyberbullying and the dissemination of inappropriate content, which can cause psychological and social harm to victims. On the other hand, smartphones can also provide a means to contact help in dangerous situations and report instances of violence. In general, smartphones can be used to perpetuate violent behaviors, although in some cases they can also be used as a tool to seek help and report violence. However, it is important for individuals to use smartphones responsibly and for authorities to take action to prevent and address violent behaviors.

Hence, it is important to note that smartphone use is not the primary cause of violence. Violence is a multifaceted phenomenon influenced by various factors, including economic and social circumstances, culture, mental health, and individual predisposition. The act of recording violence with smartphones can have various psychological aspects, both for the person recording and the victim. Regarding those recording violence, there may be several motivations. Some individuals may do it to document the incident as evidence for authorities, while others may do it to publish the video on social media to gain attention and views. Still, others may do it as a form of revenge or intimidation towards the victim [29].

Several recent studies [30–34] have investigated the ways in which groups of youths use social media to facilitate crime and violence. Among the main forms of crime identified were drug dealing, downloading music and videos illegally, and threatening, insulting, and blackmailing people through video recordings. Furthermore, Decker and Pyrooz [30] highlight the use of social media by the general youth population to communicate and flirt with peers but also to share music and videos of various kinds with their network of contacts. In the same way, so-called juvenile criminal gangs use social media for the promotion of their illegal activities, boasting about violence and threats and enhancing the group's image.

An additional study [33] which involved the participation of 585 former gang members and perpetrators of violent crimes revealed through interviews that about 45% of the sample

were involved in at least one form of online crime, including harassment, threats, and uploading violent videos to the network.

Unfortunately, there have been several reported cases involving violence and smartphone use. One of the most well-known cases is cyberbullying, where victims are attacked online through social media or other digital platforms [35,36]. Examples of cyberbullying include the sending of offensive or threatening messages, the spreading of rumors, the disclosure of personal information, the displaying of embarrassing images, or the exclusion of others during online communications [37]. Although there appears to be significant conceptual overlap between face-to-face bullying and cyberbullying [38,39], cyberbullying differs from traditional bullying in that humiliating texts or visual materials sent via social media can be permanent and publicly available [40]. Furthermore, while face-to-face bullying is generally characterized by physical dominance, in cyberbullying, a physical advantage is not necessary; perpetrators can dominate a victim through knowledge of social media usage, anonymity, limited defensive options for the victim, and few avenues of escape [41].

Online bullying can cause psychological and social harm to victims and may have negative consequences for mental health [42]. For the victim, being recorded during an act of violence can cause shock, shame, anxiety, depression, and suicidal thoughts [43]. Additionally, becoming a public figure on the internet can lead to negative social consequences, such as the spread of offensive comments or loss of privacy. In general, recording violence with smartphones can have a negative impact on the mental health and psychological well-being of those involved. It is important to consider the right to privacy and dignity of individuals and to address the issue of violence effectively, without resorting to methods that may worsen the situation. For example, we found cases of sexual violence where the aggressor recorded their actions with a smartphone and then shared the video on the internet. This behavior is considered a form of revenge porn, exposing the victim at a public level [44]. There have also been cases of domestic violence where the aggressor controlled the victim's phone, limiting their freedom and access to support resources.

According to the literature, the lack of empathy can be a contributing factor to violence [45]. Empathy is the ability to understand and share the feelings of others, and thus it is essential for building healthy and respectful relationships. Those lacking empathy may have difficulties understanding the pain and suffering of others, making them less inclined to prevent or avoid violence [46]. In some cases, lack of empathy, risk taking, and low fear of punishment may be associated with personality disorders, such as antisocial personality disorder. Furthermore, lack of empathy can also be related to other issues, such as aggressiveness, impulsivity, tendencies towards manipulation, and dishonesty, i.e., maladaptive schemas [47]. These behaviors can contribute to perpetuating situations of violence and abuse.

Limitations of this study must be also pointed out. First, the study analyzed only 80 recent cases of violence, which might not be representative of the entire spectrum of smartphone-mediated violence incidents. Second, the study discussed the relationship between smartphone use and violence but did not establish a causal link. Other confounding variables that could contribute to violent behaviors were not fully controlled.

## 5. Conclusions

To our knowledge, this is the first study focusing on technological aspects (smartphone-related violence and violent video use and motivations) providing valuable insights and a starting point for research. By identifying patterns and psychological aspects, the present study contributes to understanding the multifaceted nature of smartphone-related violence. In particular, the present study refers to the broad literature, explaining well-known mechanisms such as humiliation [48], perversion [49], and revenge [50] applied in real-world scenarios for the first time, particularly in social media and social contexts.

Moreover, the recognition of smartphones as tools for both perpetuating and combating violence emphasizes their dual role. This study underscores the importance of

responsible smartphone use, awareness, and intervention strategies to address and prevent violent behaviors.

The case studies underscore the necessity of fostering a culture of respect, empathy, and accountability. Comprehensive education on respectful behavior, both online and offline, is crucial to preventing and addressing such cases. It is a call to action for society, institutions, and individuals to collectively work towards promoting safety, well-being, and ethical use of technology, while holding perpetrators accountable for their actions and advocating for the rights and dignity of all beings affected by violent behaviors.

## 6. Notes on Violence Prevention

In the contemporary landscape where smartphones are ubiquitous, the intersection of violence and digital recording poses social, psychological, and legal issues. The capacity of smartphones to document real-time events has transformed every individual into a potential witness or reporter, which has far-reaching implications for both perpetuating and preventing violence. This essay will explore these implications, delineating the challenges posed by the recording of violence and the opportunities that arise for prevention [51].

One of the most significant challenges is the potential desensitization to violence. As violent events are recorded and circulated, the shock value diminishes, which may reduce the urgency of the response from viewers and authorities alike [52]. Moreover, the aspiration for virality can spur individuals to commit acts of violence with the express purpose of capturing and sharing them online, seeking notoriety in a digital sphere that often rewards shock and sensationalism [53].

The trauma experienced by victims of violence is amplified when recordings of their most vulnerable moments are disseminated. The permanence and inescapability of the digital footprint can lead to prolonged suffering and a sense of violation that extends beyond the initial act of violence. Legal and ethical boundaries are also at stake; the act of recording can infringe on privacy rights and, if shared, can potentially contaminate legal proceedings by influencing public opinion and juror impartiality [54].

Despite these challenges, smartphones also present opportunities to educate and foster awareness. Educational programs can emphasize responsible digital citizenship, including the ethical implications of recording and disseminating violent incidents [55]. Technological solutions can be engineered to identify and restrict the spread of violent content, thereby curtailing its virality.

Bystander intervention training presents another avenue for change. Instead of recording violence, bystanders can be equipped with the skills and knowledge to de-escalate situations or seek immediate assistance, thereby contributing to a culture of proactive support rather than passive observation [56]. Additionally, creating strong support systems for victims, including psychological counseling and legal assistance, can help mitigate the harm caused by such recordings.

The dual role of smartphones in violence—both as a medium that can exacerbate trauma and as a tool for advocacy and prevention—underscores the need for nuanced approaches to digital conduct. Through a combination of education, technology, policy, and community engagement, it is possible to leverage the power of smartphones for good, turning them from instruments that can perpetuate violence into tools for justice and support.

**Author Contributions:** Conceptualization, A.R. and E.P.; methodology, A.R.; software, A.R.; validation, E.P., G.I. and A.R.; formal analysis, A.R.; investigation, E.P.; resources, G.I.; data curation, E.P. A.R.; writing—original draft preparation, all; writing—review and editing, A.R. and G.I.; visualization, E.P.; supervision, A.R.; project administration, A.R. All authors have read and agreed to the published version of the manuscript.

**Funding:** This research received no external funding.

**Institutional Review Board Statement:** Not applicable.

**Informed Consent Statement:** Not applicable. Data anonymity was granted according to the Declaration of Helsinki regarding research on human subjects.

**Data Availability Statement:** The dataset analyzed and generated during the study can be obtained under reasonable request.

**Acknowledgments:** The authors wish to thank the fellow researchers who provided over 20 reviews and useful feedback on the Qeios platform for the development of this study.

**Conflicts of Interest:** The authors declare no conflict of interest.

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
