# Peer review of "Exploring the Link between Smartphone Use, Recorded Violence, and Social Sharing in 80 Case Studies in Italy"

_psych, doi:10.3390/psych5040082_

Round 1

Reviewer 1 Report

Comments and Suggestions for Authors

Dear Authors.

Congratulations on an interesting article. The topic itself and the analytical approach is very original. The problem of recorded violence is very topical and too rarely described in the literature. The text needs, in my opinion, only minor changes.

Title: is it not worth noting that recorded violence is studied

Abstract: please add from what period the case studies discussed were from

Introduction: the entire lengthy paragraph on lines 57-92 is unreadable because it contains points that are written one after another. I suggest correcting it.

Aim: isn't the aim also to categorize the cases, it looks like this is the result of qualitative analysis

Material and methods: please write who collected the cases and whether there were any selection criteria

Analysis: the phrase "advanced statistical methods" is unclear. In addition, section 2.2 should correspond better with the results and tables/figures.

I don't see a reference to Figure 3, which is also poorly described in the text. I am sorry but I don’t understand it (and similar axis titles)

Conclusions. Limitations are more fitting at the end of the discussion.

General comment on the figures:

-           their quality is unacceptable,

-          title inside the figure is no longer needed (in all 3 graphs),

-          I don't know why the colours were introduced (once explained).

Comments on the Quality of English Language

I don;t feel to be an expert

Author Response

REVIEW REPORT 1

Dear Reviewer,

We would like to express our sincere gratitude for the insightful comments and constructive suggestions you provided regarding our manuscript. Your thorough review and valuable feedback have significantly contributed to enhancing the quality of our paper.

Your attention to detail in identifying areas that needed clarification or improvement, and your suggestions for how to address these issues, were particularly helpful. We believe that the revisions we have made, in response to your comments, have greatly strengthened our argument and the overall presentation of our research.

Here, a point by point answer to your suggestion.

R: Congratulations on an interesting article. The topic itself and the analytical approach is very original. The problem of recorded violence is very topical and too rarely described in the literature. The text needs, in my opinion, only minor changes.

A: Thank you so much.

R: Title: is it not worth noting that recorded violence is studied

A: We have revised the title to emphasize that our study focuses on recorded violence. This change makes the title more reflective of the content of our paper.

R: Abstract: please add from what period the case studies discussed were from

A: In the abstract, we have added information about the time period from which our case studies are drawn. This provides readers with essential context right at the outset.

R: Introduction: the entire lengthy paragraph on lines 57-92 is unreadable because it contains points that are written one after another. I suggest correcting it.

A: We restructured the problematic paragraph in the introduction (lines 57-92) for better readability. We organized the points more coherently, ensuring the flow of information is clear and logical.

R: Aim: isn't the aim also to categorize the cases, it looks like this is the result of qualitative analysis

A: We have updated the aim of our study to explicitly include the categorization of cases. This amendment reflects the outcomes of our qualitative analysis more accurately.

R: Material and methods: please write who collected the cases and whether there were any selection criteria

A: In the 'Materials and Methods' section, we have included details about who collected the case studies and the selection criteria applied. This addition offers greater transparency and understanding of our research process. Thank you.

R: Analysis: the phrase "advanced statistical methods" is unclear. In addition, section 2.2 should correspond better with the results and tables/figures.

A: We clarified the use of the term "advanced statistical methods" by providing more specific details. Additionally, we aligned section 2.2 more closely with our results and the tables/figures presented, ensuring consistency across the paper.

R: I don't see a reference to Figure 3, which is also poorly described in the text. I am sorry but I don’t understand it (and similar axis titles)

A: We have added a reference to Figure 3 in the relevant section and improved its description in the text. We ensured that the axis titles and the graph's purpose are now clearly understandable, addressing the previously noted confusion.

R: Conclusions. Limitations are more fitting at the end of the discussion.

A: We moved the discussion of limitations to the end of the discussion section. This repositioning aligns with the standard scientific reporting format and helps readers better understand the scope and constraints of our study.

R: General comment on the figures:

-           their quality is unacceptable,

-          title inside the figure is no longer needed (in all 3 graphs),

-          I don't know why the colours were introduced (once explained).

A: We have addressed the concerns about the quality of the figures. We removed the titles inside all three figures for a cleaner presentation and provided an explanation for the color scheme used, ensuring that each figure is both visually appealing and informative.

We deeply appreciate the time and effort you have dedicated to reviewing our work. Your expertise and thoughtful critique have been instrumental in guiding our revisions and helping us present our research more effectively.

The corresponding author

On behalf of all co-authors

Reviewer 2 Report

Comments and Suggestions for Authors

This study investigated the relationship between smartphone use, violence, and social sharing using a qualitative method to analyze news outlets. In general, it is a good start to explore the issues of sharing violence act or information in social media. The motivations identified by the qualitative analysis can provide insight for future studies. However, the authors can provide further explanation and references to support these motivations and the definition they stated. The authors could discuss further on the practical implications and future directions. The graphs in the manuscript have low resolution which affect readability, please improve.

Author Response

REVIEW REPORT 2

Dear Reviewer,

We appreciate your recognition of the study as a good starting point for exploring the issues of violence and its dissemination through social media. We agree that the motivations identified in our qualitative analysis are crucial for future research in this area.

In response to your suggestion, we expanded the explanation of the motivations identified in our study. We plan to include additional references and literature to support and contextualize these motivations, thereby strengthening the foundation and credibility of our analysis.

We acknowledge the importance of discussing the practical implications of our findings. In the revised manuscript, we included a detailed section discussing how our findings can be applied in real-world scenarios, particularly in media and social policy contexts. Additionally, we will outline potential directions for future research, highlighting how our study paves the way for further exploration into the dynamics of smartphone-mediated violence.

We understand the importance of clear and legible graphical representations in enhancing the readability and overall quality of the manuscript. We will address the issue of low-resolution graphs by either recreating them with higher resolution or employing more effective graphical tools. This will ensure that the revised graphs are both aesthetically pleasing and effectively convey the intended data.

Thank you again for your constructive feedback, which is invaluable in helping us refine and improve our study. We are committed to addressing these points thoroughly in our revised manuscript.

The corresponding author on behalf of all co-authors